# Application of an Expandable Cage for Reconstruction of the Cervical Spine in a Consecutive Series of Eighty-Six Patients

**DOI:** 10.3390/medicina56120642

**Published:** 2020-11-25

**Authors:** Mirza Pojskic, Benjamin Saβ, Christopher Nimsky, Barbara Carl

**Affiliations:** 1Department of Neurosurgery, University of Marburg, 35043 Marburg, Germany; sassb@uni-marburg.de (B.S.); nimsky@med.uni-marburg.de (C.N.); Barbara.Carl@helios-gesundheit.de (B.C.); 2Marburg Center for Mind, Brain and Behavior (MCMBB), 35032 Marburg, Germany; 3Department of Neurosurgery, Helios Dr. Horst Schmidt Kliniken, 65199 Wiesbaden, Germany

**Keywords:** expandable cage, cervical spine, corpectomy, anterior spinal column

## Abstract

*Background and objectives*: Expandable cages are frequently used to reconstruct the anterior spinal column after a corpectomy. In this retrospective study, we evaluated the perioperative advantages and disadvantages of corpectomy reconstruction with an expandable cage. *Materials and Methods:* Eighty-six patients (45 male and 41 female patients, medium age of 61.3 years) were treated with an expandable titanium cage for a variety of indications from January 2012 to December 2019 and analyzed retrospectively. The mean follow-up was 30.7 months. Outcome was measured by clinical examination and visual analogue scale (VAS); myelopathy was classified according to the EMS (European Myelopathy Scale) and gait disturbances with the Nurick score. Radiographic analysis comprised measurement of fusion, subsidence and the C2–C7 angle. *Results*: Indications included spinal canal stenosis with myelopathy (46 or 53.5%), metastasis (24 or 27.9%), spondylodiscitis (12 or 14%), and fracture (4 or 4.6%). In 39 patients (45.3%), additional dorsal stabilization (360° fusion) was performed. In 13 patients, hardware failure occurred, and in 8 patients, adjacent segment disease occurred. Improvement of pain symptoms, myelopathy, and gait following surgery were statistically significant (*p* < 0.05), with a medium preoperative VAS of 8, a postoperative score of 3.2, and medium EMS scores of 11.3 preoperatively vs. 14.3 postoperatively. Radiographic analysis showed successful fusion in 74 patients (86%). As shown in previous studies, correction of the C2–C7 angle did not correlate with improvement of neurological symptoms. *Conclusion*: Our results show that expandable titanium cages are a safe and useful tool in anterior cervical corpectomies for providing adequate anterior column support and stability.

## 1. Introduction

Expandable cages (ECs) are frequently used to reconstruct the anterior spinal column after corpectomy. Indications include spinal canal stenosis with compression of the spinal cord [1], fracture [2,3], spondylodiscitis [4], and metastases [5]. Corpectomy of the cervical spine is performed through the anterior approach with the need of a graft or implant device for reconstruction [6]. ECs can be adjusted to the size of corpectomy [7], and they offer solid anterior support, providing excellent primary stability [6] with less risk of damage to the end plate, less intraoperative manipulation of the device, and potentially greater control over lordosis in comparison to non-expandable cages [8]. They may be particularly advantageous in cases with poor bone quality, such as patients with osteoporosis or metastatic tumors that have been radiated [8]. However, there is a potential risk of overdistraction, and the amount of hardware in the expansion mechanism may limit the surface area available for fusion [8]. The goal of our study was to investigate the potential advantages and disadvantages of corpectomy reconstruction in the cervical spine with a distractable cage. To our knowledge, this is the largest study on the implantation of ECs for reconstruction of the anterior spinal column.

## 2. Materials and Methods

Eighty-six patients were treated with an expandable titanium cage (X-Core Mini, NuVasive, San Diego, CA, USA) for a variety of indications from 2012–2019 and analyzed retrospectively. Data were gathered through review of patient electronic records and relevant imaging. Indications for standard anterior corpectomy and reconstruction with expandable cage included spinal canal stenosis with myelopathy and ventral compression of the spinal cord, neurological deficits with pyramidal tract signs, or presence of kyphotic deformity. Further indications were spondylodiscitis with sepsis and empyema, where due to the compression on the spinal cord and damaged bony substance at risk of a fracture conservative treatment was not a reasonable option. Metastases with intractable pain which occupied more than 50% of the vertebral body with danger of development of pathological fracture and spinal cord compression, which were due to pain and development of neurological deficits not suitable for conservative non-operative treatment, i.e., radiotherapy, were also candidates for corpectomy and expandable cage implantation. Indications also included instable multifragmentary vertebral body fracture with instability of anterior column, intractable pain, neurological deficits and compression of the spinal cord. In cases of instability, additional instrumentation surgery was performed.

In all cases, gadolinium contrast-enhancing magnetic resonance imaging (MRI) of the spine as well as computed tomography (CT) of the spine were obtained. All patients received CT and X-ray of the instrumented region on the first day following surgery. Neuroimaging was verified by independent neuroradiologists.

Indications for single-level corpectomy in patients with degenerative spine disease included spinal canal stenosis with myelopathy due to herniated discs on the adjacent levels and ventral compression of the spinal cord from posterior aspect of the vertebral body due to osteophyte formation where it was not expected that the posterior decompression alone, with or without instrumentation would provide sufficient decompression. Two-level cervical discectomy and fusion, with or without the posterior approach, was discussed with patients as a less invasive therapy alternative. Conservative treatment was due to myelopathy and neurological deficits not being an option. In cases of metastases and spondylodiscitis, one-level corpectomy was performed when the vertebral body was infiltrated by the tumor or infection at an extent of more than 50% or in the presence of empyema along the posterior aspect of the vertebral body.

In patients with fracture, multifragmentary fracture of the vertebral body where the instability of the anterior column was not suitable for stabilization via the posterior approach alone was an indication for multilevel corpectomy.

Indications for multilevel corpectomy in patients with degenerative spine disease included severe spinal canal stenosis with myelopathy due to herniated discs and osteophyte compression on the spinal cord from the posterior aspect of the vertebra in more than one level, where we did not expect that the posterior decompression alone, with or without instrumentation, would provide sufficient decompression of the spinal cord. In multilevel spinal canal stenosis with a significant amount of compression on the spinal cord due to posterior vertebral body osteophytes, corpectomy was chosen over multilevel anterior cervical discectomy and fusion. Multilevel anterior cervical discectomy and fusion, with or without the posterior approach, was discussed with patients as a less invasive therapy alternative. In cases of metastases, we performed multilevel corpectomy of vertebral bodies which were infiltrated by the tumor at an extent of more than 50%. In spondylodiscitis, multilevel corpectomy was performed in cases of profound involvement of more than one vertebral body and adjacent discs, as well as in cases of the presence of epidural empyema along the posterior aspect of the vertebral body. In patients with fracture, multifragmentary fracture of two adjacent vertebral bodies where the instability of the anterior column was not suitable for stabilization via the posterior approach alone was an indication for multilevel corpectomy.

Instability of the cervical spine was postulated when two or more columns were encased in the underlying pathology or in the case of dorsal compression on the spinal cord. On translation motion X-ray and CT scan of the cervical spine, cervical instability is considered present if the horizontal displacement of cervical vertebra is more than 1.7 mm or if the angular displacement is more than 5.7°. Additional dorsal stabilization with lateral mass screws and/or pedicle screws was performed.

Anterior cervical plating was applied in all patients where anterior column support was not sufficiently achieved by corpectomy and implantation of the expandable cage alone, i.e., in cases of kyphotic instability of the anterior spinal column, as well as in cases where additional posterior stabilization was not performed. Standalone constructs (corpectomy with EC implantation without plate and without posterior stabilization) were performed only in cases of single- and two-level corpectomy without signs of instability and restored sagittal alignment. In all other cases of single- and multiple-level corpectomy without posterior stabilization, additional plating was performed. In cases of 360° fusion, plating was performed for multilevel constructs as well as in cases of pathologically changed bony substance (metastases, fracture and spondylodiscitis). Variable angle screws were used with the goal of bicortical purchase.

Follow-up comprised clinical examinations, assessment of the visual analogue scale (VAS), EMS (European Myelopathy Scale) and the Nurick score at 3, 12, and 24 months after surgery.

X-ray scans were obtained at 3, 12 and 24 months following surgery, and dynamic flexion–extension X-ray 6 months following surgery, while CT scans were obtained in the period of 6–24 months. Radiographic analysis comprised measurement of fusion, subsidence, and the Cobb C2–C7 angle.

Fusion was defined as the presence of trabeculae bridging bone formation at the anterior and/or posterior cortex of the involved vertebral bodies on the CT scan, and at the interface between the cage and the vertebral endplate. Absence of such bridges was classified as non-fusion.

Subsidence was considered to have occurred if either the anterior or posterior height of the intervertebral space decreased by more than 1 mm from that measured on the postoperative radiograph due to sintering of the implant in the endplates of the adjacent vertebra [9,10].

Cervical lordosis was determined using the Cobb C2–C7 angle with the four line method, which included drawing a line parallel to the inferior endplate of C2 and C7 with perpendicular lines drawn from the each of the two lines; the angle resulting from crossing of the perpendicular lines is defined as the Cobb C2–C7 angle [11] (Figure 1).

The analyses was performed using SPSS statistical software, version 20 (SPSS Inc., Chicago, IL, USA). A *p* value < 0.05 was considered to be statistically significant.

## 3. Results

Eighty-six patients underwent corpectomy and anterior column reconstruction with ECs for spinal canal stenosis with myelopathy (Figure 2 and Figure 3), metastasis, spondylodiscitis, and fracture. Four patients (4.65%) died in the early postoperative period of less than three months, thus, they were excluded from the outcome assessment. Two patients with spondylodiscitis died due to sepsis, one patient with metastasis due to cancer progression and one patient with fracture due to pulmonary embolism. General characteristics of the patients are summarized in Table 1.

### 3.1. Clinical Outcome

Pain reduction showed statistical significance with a preoperative mean VAS of 8 and a postoperative mean VAS of 3.2 (paired samples *t*-test, *t* = 16.975, corr = 0.483, *p* < 0.05).

Seventy-two or 83.7% of patients had a neurological deficit prior to surgery. C5 and C6 palsy and spinal ataxia were the most common deficits. Four of these patients died in the early postoperative period. Out of four patients who died, three patients remained neurologically unchanged, and one patient worsened following surgery. This patient underwent dorsoventral surgery for C7 fracture and Bechterew disease and developed paraplegia due to a postoperative hematoma following dorsal stabilization and decompression.

From 82 patients included in the follow-up, 37 patients or 45.2% were postoperative neurologically unchanged, 39 (47.5%) improved and 6 (7.3%) worsened. From these six patients, two underwent dorsoventral surgery and four multilevel ventral surgery only. All six patients had neurological decline due to postoperative worsening of C5 palsy. Two patients underwent revision surgery for hardware failure, and four patients were treated conservatively. In four patients (two who underwent revision and two who were treated conservatively), C5 palsy improved in the follow-up.

**Neurological improvement** was significant (Pearson’s coefficient chi-square = 28.064, *p* < 0.05) and was also shown by a significant increase in the EMS Score from 11.26 preoperative to 14.34 postoperative (*t* = −10.021, *p* < 0.05, corr = 0.661) as well as significant decline in Nurick score (preoperative 3.14 vs. 2.18 postoperative, *t* = 7.709, *p* < 0.05, corr = 0.702). Neurological improvement rate (chi-square = 14.7, *p* < 0.05) and EMS score postoperatively were significantly higher in patients with spinal canal stenosis compared to the other two groups (*t* = −4.06 spondylodiscitis, *t* = −3.06 metastases; Figure 4). Length of follow-up did not show a correlation with neurological improvement.

### 3.2. Radiological Outcome

Fusion occurred in 74 patients (86%) (Figure 5). Subsidence occurred in 20 patients (24.4%) (Figure 6). Subsidence occurred in 10 out 46 patients with spinal canal stenosis to follow-up, in 3 out of 10 patients with spondylodiscitis, in 6 out of 23 patients with metastases, and in 1 out of 3 patients with fracture. There were no statistically significant differences in occurrence of subsidence between the different indication groups as well as between monosegmental und multisegmental surgery.

Cobb C2–C7 angle was preoperative 14.1° (0–78.1) vs. postoperative 12.3° (3–38.6), but this difference was not significant. Correction of lordosis was defined as difference of postoperative and preoperative Cobb C2–C7 angle less than 0° and was achieved in 36 (43.9%) patients. Correction of lordosis did not correlate with neurological improvement.

### 3.3. Complications

In 17 patients (19.8%), complications occurred in the early postoperative period of less than three months following primary surgery. In 13 patients (15.1%), hardware failure with dislocation of the expandable cage occurred during the first three weeks following surgery (7 patients) or during the postoperative period of more than three weeks but less than three months (6 patients). From these 13 patients, there were 5 patients with spinal canal stenosis, 4 with metastases, 3 with spondylodiscitis and 1 with fracture. In seven patients, hardware failure occurred following ventral and in six following dorsoventral primary surgery. In nine cases, multilevel construct was revised, and in four cases, single-level construct. In six cases, revision surgery was performed via a ventral approach; three patients received additional dorsal stabilization. Moreover, in three patients, dorsoventral revision was performed, one patient rejected revision surgery, and one patient was treated conservatively. Further complications include dorsal wound revision (three patients), dorsal screw malposition or breakage (2), esophagus injury (two patients), prevertebral hematoma (one patient), dorsal hematoma (1) and laryngeal nerve palsy (three patients). Eight patients experienced postoperative dysphagia which resolved in all patients. The first surgery with EC implantation at our department was performed in 2012. Complications occurred in 12 out of 50 patients who underwent surgery during the first 42 months following introduction of this surgical technique at our department. Furthermore, 9 out of 13 hardware failures occurred during this period, which shows a certain learning curve to address technical problems.

In patients with cervical spinal canal stenosis, five patients experienced hardware failure, and from this number, in two patients, adjacent segment disease with fracture of adjacent vertebral body occurred. Further complications were esophagus injury (two patients) as well as dorsal wound revision (one patient) and dorsal screw revision (two patients). Three patients with spondylodiscitis experienced hardware failure; there were no further non-EC-related complications in this group. In four patients with metastases, hardware failure with adjacent segment disease occurred; in two patients, wound revision was performed; and in one patient, prevertebral hematoma was evacuated. One patient with fracture had hardware failure. One patient each with spinal canal stenosis, spondylodiscitis and metastasis had a laryngeal nerve palsy following surgery.

Out of 13 patients with hardware failure, in 9 cases, adjacent segment disease (ASD) occurred (8.8%) with fracture of the proximal (*n* = 4) or distal (*n* = 5) adjacent vertebral body (4 with metastases, 2 with spinal canal stenosis, 2 spondylodiscitis, and 1 patient with fracture and Bechterew disease). In four further hardware failure cases, simple dislocation without fracture occurred (Figure 7). In seven patients, revision surgeries were performed during the same hospital stay. In one case, the surgery was declined, and in five further cases, revision surgery was performed in the early postoperative period of less than three months following primary surgery. In the long term, one patient developed secondary instability in the cervical spine 72 months following the surgery, which led to dorsal stabilization (Figure 8). 

Patients who underwent dorsoventral surgery had significantly more frequent complications compared to ventral surgery alone (chi-square = 9.236, *p* < 0.05). Patients who underwent one-level surgery had more favorable neurological outcome than patients with multilevel surgery (chi-square = 12.623, *p* < 0.05). Patients with single-level surgery had lower hardware failure rate (8.5%) than patients with multiple-level surgery (18.2%; *p* < 0.05). There were no statistically significant differences in terms of neurological and radiological outcome between patients with ventral and dorsoventral surgery (*p* < 0.05). Furthermore, there were no statistically significant differences in terms of neurological and radiological outcome as well as complication rate between patients with and without plates (*p* < 0.05). Patients with standalone ECs without a plate (*n* = 13) experienced no perioperative complications, whereas patients with standalone ECs with a plate (*n* = 34) experienced four cases of hardware failure.

## 4. Discussion

Expandable titanium cages were developed to facilitate the implantation of the device and to enhance its contact with the endplates; furthermore, implantation allows a slight distraction during the procedure to reduce a kyphotic deformation [12,13]. Cadaver studies have shown that ECs in comparison to a tricortical iliac crest bone graft and a nonexpandable cage have no biomechanical advantages [14]. Widespread use following all corpectomies is not justified due to their significantly greater cost compared to structural bone grafts or non-expandable cages [8].

Several clinical studies have shown the efficacy of the expandable cage, mostly in the cervical spine [2,3,4,5,6,7,10,15,16,17,18,19,20,21]. Most common indications were, similar to our study, cervical spinal canal stenosis [7,12], metastases [5,10,17], fractures [2,6] and spondylodiscitis [4,19]. Indications for surgery of spine metastases include intractable pain, spinal cord compression and stabilization of impending pathological fractures [5]. The use of metal ECs in cases of spondylodiscitis has shown that they maintain alignment while not perpetuating infection [4].

Pain reduction in our study was shown to be significant, which is in concordance with the published series [1,5,12,13,21]. Neurological outcome was favorable in 93.1% of patients who improved or remained unchanged following surgery. Literature review reveals an unfavorable outcome in 3.9% [12] to 10% [7]. Waschke et al. describe 85% of patients who neurologically improved at the follow-up [1]. The main reason for worsened postoperative neurological status was a C5 palsy. This is a well-known phenomenon following cervical spine surgery described as a result of anterior and posterior decompression [22] with reported incidence of 4% following corpectomy and 1% following anterior cervical discectomy and fusion [23] and a reported higher incidence following multilevel corpectomy [24]. Following ventral surgery, C5 palsy was found to be associated with cervical lordosis correction, pre-existent C4/5 foraminal stenosis, and larger extend of decompression [25]. Palsy might be also caused by C5 nerve root traction in posterior decompression surgery [22] as a result of dorsal spinal cord shift under the precondition of a foraminal stenosis [26].

Patients who underwent one-level surgery had a more favorable neurological outcome. In EC reconstruction used for more than one level, the corpectomy complication rate significantly increases [16]. In our study, the overall complication rate did not differ between the patients with single- and multiple-level reconstructions; however, hardware failure rate with adjacent segment disease occurred more frequently following multiple-level surgery.

High fusion rate is in concordance with previous studies [7]. Although several studies reported a fusion rate of 100% at the follow-up [5,7,13,27], it is important to note that these studies had shorter mean follow-up and that the techniques of fusion assessment were different, as these authors used non-movement of the EC on flexion–extension X-ray as a fusion assessment tool. Waschke et al. showed in their study of 48 patients with ECs in the cervical spine with mean follow-up of 23 months that successful fusion was seen in 79% controlled by flexion–extension X-ray and only in 48% of patients controlled by CT (with fusion being defined as the presence of bony trabeculae) [1]. Radiological outcome, i.e., fusion and correction of lordosis, did not show a correlation to neurological outcome [1]. ECs carry a potential risk of overdistraction, which is increased in the cervical spine, their minimum height limits their use in cases with collapsed vertebra, and the amount of hardware in the expansion mechanism may limit the surface area available for fusion [8].

Subsidence of the EC was described to be a very frequent event before a successful fusion was achieved due to the stiffness of titanium in comparison to normal bone [13], but as the extent of subsidence is very small, it did not cause a loss of correction [10,13]. For instance, subsidence rate was described to reach up to 17% and showed no correlation to pathology or EC level [10]. ECs were independently associated with higher rates and odds of subsidence in comparison to static cages and with a 1-year subsidence rate of 51.6% [18]. In our study, subsidence occurred in 24.4% of patients to follow-up. Most hardware-related complications such as subsidence were found not to be symptomatic and can be treated conservatively [28]. Interestingly, we did not find any correlation between cage subsidence and different pathologies of the cervical spine in regard to the quality of the bone substance. A recent study by Lau et al. revealed some interesting aspects on subsidence in relation to ECs [18], which was found to be associated with infection but also with trauma and younger age of patients. 

In 17 patients (19.8%), perioperative complications occurred. This number is in concordance with previous studies where up to 24% of patients require a revision surgery [16]. Patients who underwent dorsoventral surgery had significantly more frequent complications compared to ventral surgery alone. The number of corpectomy levels and the surgical approach have been found to correlate with the risk of complications, especially those non-hardware related [16]. The greater the number of fusion segments, the greater the incidence of complications [28].

The term “adjacent segment degeneration” describes radiographic changes seen at levels adjacent to a previous spinal fusion procedure that do not necessarily correlate with any clinical findings. The term “adjacent segment disease” (ASD) refers to the development of new clinical symptoms that correspond to radiographic changes adjacent to the level of a previous spinal fusion [29]. ASD occurred in nine patients (8.8%),which is above the reported range of 3.15% [10] to 3.9% [12] in EC surgery. However, recent meta-analysis has shown that pooled adjacent segment degeneration following cervical fusion surgery was 32.8% in one-quarter to one-third of the patients developing ASD [30]. Over distraction of the EC and poor bone quality due to metastases or osteoporosis have been found to be risk factors for ASD [31], as well as multilevel corpectomy [32]. The fracture pattern in the coronal plane appears to be similar in all cases of ASD [31]. Seven out of nine patients with ASD in our study had poor bone quality. Hardware failure in the EC has been shown to be less common than reconstruction with non-EC where early failure rate increases to 67% [33]. The etiology of ASD is multifactorial and most cases are found to be unavoidable. Evaluation of possible nonfusion alternatives and proper selection of corpectomy levels with attempt to restore sagittal alignment are needed when patients have a high risk of developing ASD [34].

Forty-seven patients received standalone cage, and from this number, 13 patients (6 one-level and 7 two-level corpectomy) received no additional plate. Furthermore, in 28 patients with 360° fusion, no additional plating was performed. Patients with and without plates had no outcome differences, and in 13 patients with standalone ECs without a plate, there were no perioperative hardware-related complications. Due to concerns regarding postoperative stability, loss of lordosis, and subsidence or migration of the implant cages, they are commonly used with supplemental fixation such as pedicle screw systems or anterior plates [35]. Anterior plates are also commonly used to stabilize corpectomy constructs [36]. With one- and two-level corpectomy constructs, the incidence of graft displacement or dislocation is low, with and without plating [37]. Although the addition of an anterior cervical plate was intended to reduce the incidence of graft-related complications, clinical series of plated multilevel corpectomies have actually been associated with higher graft complication rates than the unplated counterparts [36,38]. Biomechanical studies have not shown that any additional stability results from adding a ventral plate once dorsal instrumentation has been performed [39]. Situations such as postlaminectomy kyphosis, osteoporosis, oncologic reconstructions, and severe deformity still may be indications for ventral cervical plating after corpectomy [40]. In cases of one- and two-level corpectomy, we adjust the decision of additional plating or dorsal stabilization tailored to individual patient characteristics such as overall stability, bone quality and underlying pathology.

The limitation of our study is its retrospective nature; however, prospective studies are needed to rigorously evaluate the efficiency of the expandable cage with long-term assessment of fusion and subsidence. Disadvantage in the evaluation of our results, especially for the group of patients of the degenerative disease, i.e., spinal canal stenosis, is that we did not have a control group of patients with this diagnosis who were treated with different surgical approaches (corpectomy and implantation of tricortical iliac crest graft, corpectomy and implantation of non-expandable cervical cage as well as discectomy in two or more levels with implantation of intervertebral cage). Further limitations of the present study lie in the variety of indications and the different pathologies of the surgically treated patients, different cervical levels treated and the different follow-up period. However, this review of our initial experience serves as a solid basis for further studies on the application of the expandable cage in certain indications. Nonhomogeneous samples of patients with different pathologies treated with corpectomy and EC implantation require careful interpretation of our results due to the variety of indications for surgery and differences in the underlying disease and quality of bone substance; however, further separate prospective studies in regard to single pathology with a control group treated with a different surgical approach are needed. 

## 5. Conclusions

Our results show that expandable titanium cages are a safe and useful tool in anterior cervical corpectomies for providing adequate anterior column support and stability. Pain and functional score improvement in most of the patients with different indications recommends the use of an expandable cage. Our initial experience and patient outcomes require careful interpretation due to the variety of indications for surgery. There is a need for prospective studies which assess the advantages and disadvantages of the use of expandable titanium cages for specific indications due to the fact that the different quality of the bone substance in patients with degenerative spine disease, tumor, fracture and infection probably substantially influences the outcome. Single-level surgery showed more favorable clinical outcomes. Due to possible occurrence of adjacent segment disease in patients with high risk, indications should be thoroughly reviewed and a continuous postoperative follow-up is recommended. 

## Figures and Tables

**Figure 1 medicina-56-00642-f001:**
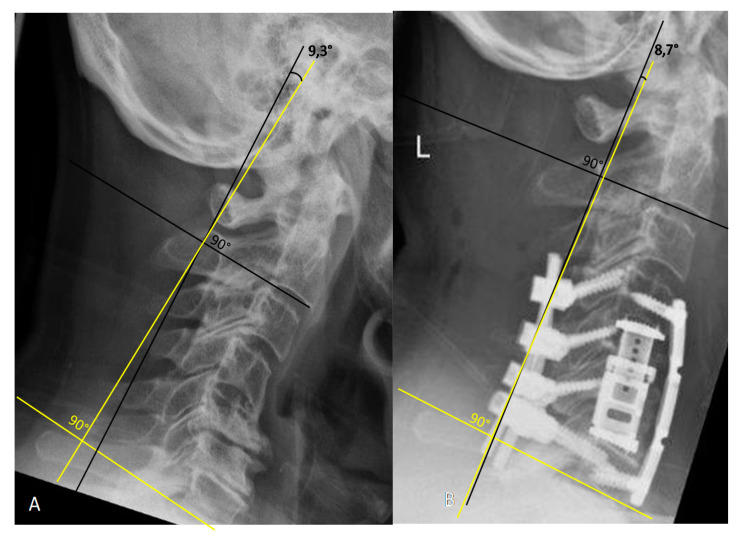
Cobb technique for measurement of the C2–C7 angle. (**A**). Preoperative lateral X-ray of the cervical spine shows kyphotic deformity with a C2–C7 angle of 9.3°. (**B**). Postoperative lateral X-ray of the cervical spine following C5/6 corpectomy with plating and dorsal stabilization C4–7 shows lordosis correction with a C2–C7 angle of 8.7°.

**Figure 2 medicina-56-00642-f002:**
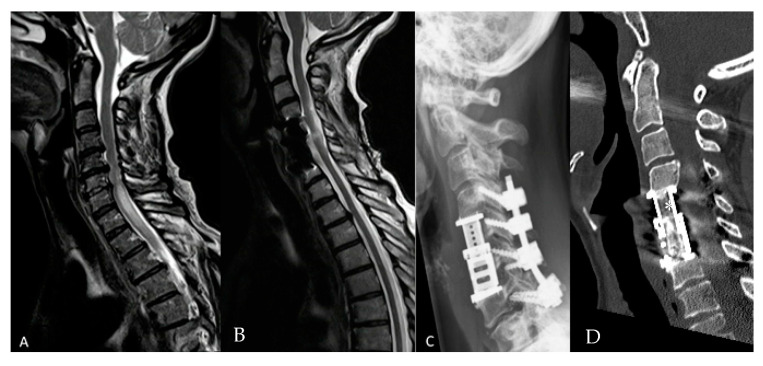
Fifty-six-year-old female patient with cervical spinal canal stenosis and myelopathy. Corpectomy with expandable cage (EC) implantation was performed due to ventral compression on the spinal cord from herniated discs and osteophytes in the level C5/6 and C6/7 with additional posterior decompression and stabilization due to instability. The posterior approach alone was discussed with a patient as a therapy alternative. (**A**) Preoperative sagittal T2 magnetic resonance imaging (MRI) of the cervical spine with spinal canal stenosis at C5/6 with myelopathy. (**B**) Postoperative sagittal T2 MRI of the cervical spine shows the resolution of myelopathy 1 year following surgery. (**C**) Postoperative lateral cervical spine X-ray shows the implant and screws following corpectomy C5/6, EC implantation and stabilization C4–7. (**D**) Postoperative computed tomography (CT) of the cervical spine shows fusion 12 months following surgery (*).

**Figure 3 medicina-56-00642-f003:**
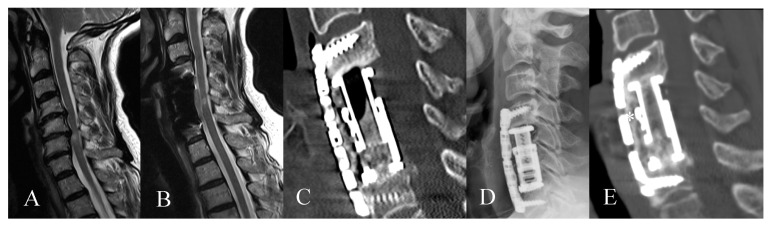
Sixty-seven-year-old female patient with cervical spinal canal stenosis C5/C6 with myelopathy. (**A**) Preoperative sagittal T2 MRI of the cervical spine with spinal canal stenosis at C5/6 with myelopathy (**B**) Sagittal T2 MRI of the cervical spine shows decompression of the spinal canal with myelopathy 1 year following surgery. (**C**) Postoperative CT of the cervical spine and (**D**) postoperative lateral X-ray of the cervical spine on the first day following the surgery following corpectomy C5/6, EC implantation and plating. (**E**) Postoperative CT of the cervical spine shows fusion 18 months following surgery (*).

**Figure 4 medicina-56-00642-f004:**
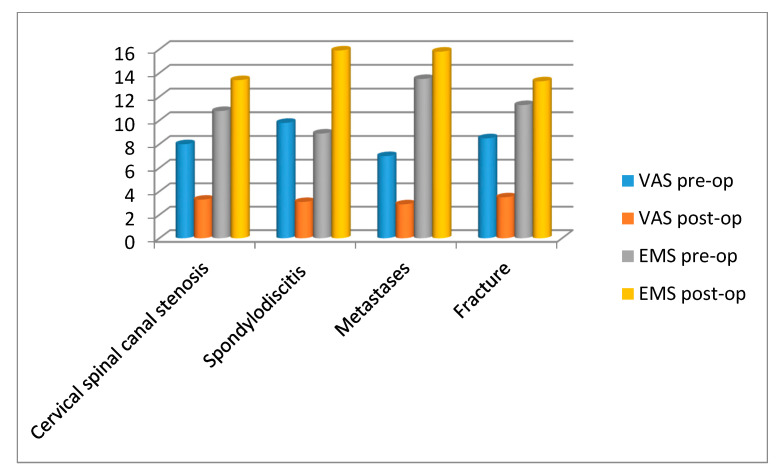
Clinical outcome parameters (visual analogue scale (VAS) and European Myelopathy Scale (EMS)) pre- and post-operatively in different pathologies (cervical spinal canal stenosis, spondylodiscitis, metastases and fracture).

**Figure 5 medicina-56-00642-f005:**
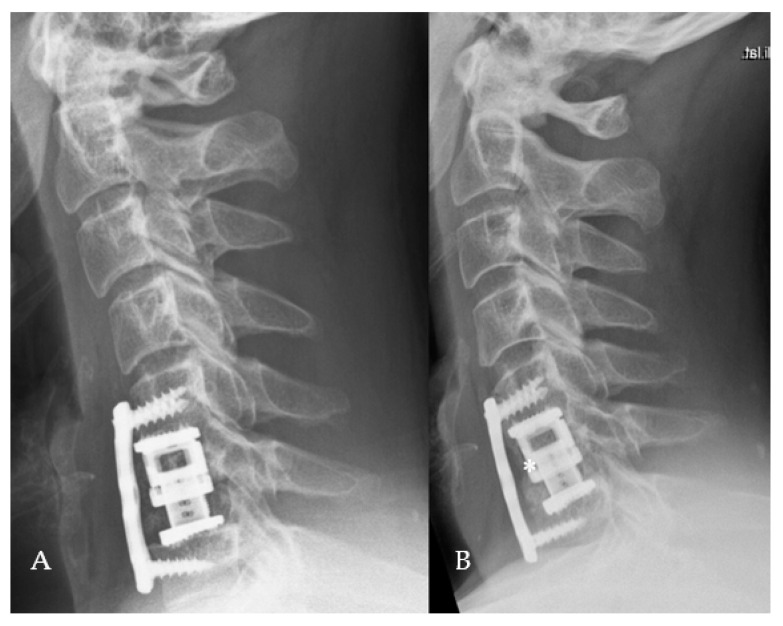
Forty-year-old patient with cervical spinal canal stenosis, corpectomy C6, EC implantation and plating. (**A**) Postoperative lateral X-ray of the cervical spine. (**B**) Lateral X-ray of the cervical spine shows fusion 3 years following the surgery (*).

**Figure 6 medicina-56-00642-f006:**
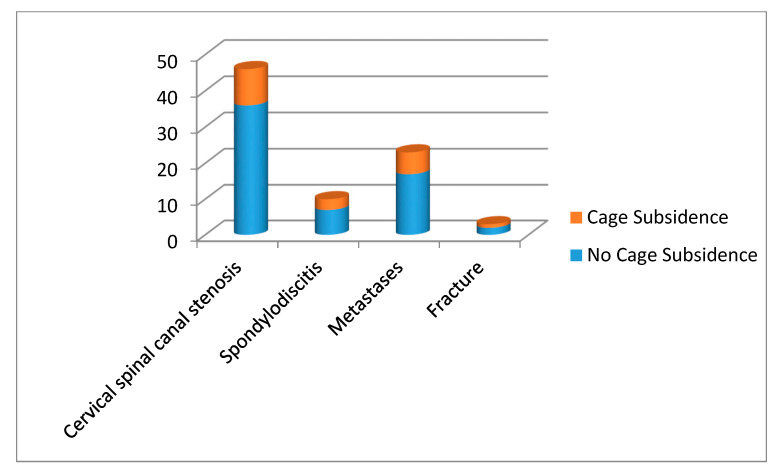
Radiological outcome parameter cage subsidence with numerical ratio of patients with and without cage subsidence between different indication groups.

**Figure 7 medicina-56-00642-f007:**
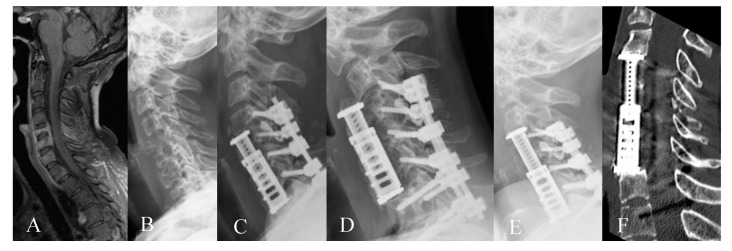
Thirty-seven-year-old female patient with multiple instable pathological fractures of the cervical spine due to metastases of breast carcinoma. (**A**) Preoperative post-contrast sagittal MRI of the cervical spine and (**B**) preoperative lateral X-ray of the cervical spine show osteolytic metastases in C5, C6, and C7. The patient underwent resection of metastases, corpectomy, EC implantation as well as dorsal stabilization 4–7. (**C**) Lateral X-ray of the cervical spine one day following the surgery. (**D**) Lateral X-ray of the cervical spine 4 weeks following surgery shows dislocation of the implant with fracture of C4 due to tumor progression. The patient underwent ventral revision with C4 corpectomy. (**E**) Postoperative lateral X-ray of the cervical spine following revision surgery. (**F**) Postoperative CT of the cervical spine one year following surgery shows no signs of dislocation.

**Figure 8 medicina-56-00642-f008:**
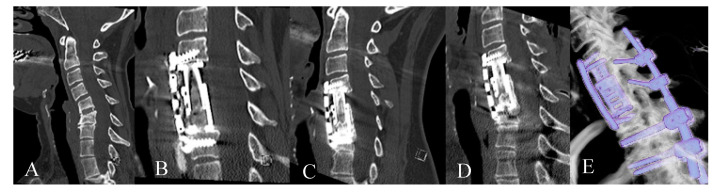
Forty-eight-year-old patient with cervical spinal canal stenosis and instability following anterior discectomy and fusion C5/6. (**A**). Preoperative CT of the cervical spine shows segmental instability C5/6. (**B**). Postoperative CT of the cervical spine following corpectomy, EC implantation and plating. (**C**). CT of the cervical spine 3 years following surgery shows fusion; subsidence at the caudal end of the construct (**D**). CT of the cervical spine 5 years following surgery shows signs of instability C6/7 which was treated with dorsal stabilization. (**E**) Reconstruction of spinal construct from CT following posterior surgery.

**Table 1 medicina-56-00642-t001:** General characteristics of the patients.

Patient Characteristics	Number
Gender	
Male	45 (52.3%)
Female	41 (47.7%)
Mean age	61.3 years (11–89)
Mean follow-up	30.7 months
Mean surgery time	191 min (67–447)
Mean hospital stay	10.6 days (3–53)
Indications	
Spinal canal stenosis with myelopathy	46 (53.5%)
Metastasis	24 (27.9%)
Spondylodiscitis	12 (14%)
Fracture	4 (4.6%)
Operative therapy at primary surgery	
Standalone	47 (54.6%)
Without plate	13
With plate	34
360° fusion	39 (45.4%)
Without plate	28
With plate	11
Corpectomy level at primary surgery	
Single level	34 (39.5%)
C2	1
C3	3
C4	5
C5	9
C6	11
C7	5
Standalone	14
Without plate	5
With plate	9
360° fusion	20
Without plate	12
With plate	8
Multiple level	52 (60.5%)
C3 and C4	2
C4 and C5	13
C5 and C6	24
C6 and C7	5
C7 and Th1	2
C4,5,6	2
C5,6,7	1
C4,5,6,7	1
C5,6,7, Th1	1
C7, Th1, Th2	1
Standalone	33
Without plate	8
With plate	25
360° fusion	19
Without plate	16
With plate	3
Levels of dorsal stabilization at primary surgery	39
C0-C5	1
C1–7	1
C2–4	1
C2–5	2
C2–6	1
C3–5	1
C3–6	7
C3–7	3
C4–6	3
C4–7	3
C5–7	3
C2-Th1	1
C3-Th1	1
C3-Th2	1
C3-Th3	2
C4-Th1	1
C4-Th4	2
C5-Th2	1
C6-Th1	1
Th2–8	1
Th6–10	1
Th3–12	1

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
