# Peer review of "Application of an Expandable Cage for Reconstruction of the Cervical Spine in a Consecutive Series of Eighty-Six Patients"

_medicina, 2020, doi:10.3390/medicina56120642_

Round 1
Reviewer 1 Report
This is an interesting study describing surgical outcomes for titanium expandable cage usage in cervical spine surgery. All disease states from degenerative to neoplasm to infection are included. This clouds the conclsuons as each of these pathologies are different, likely with differing degrees of "normal" bone upon which the cage is expanded.
From the examples provided by the authors, the usage of multilevel corpectomy seems excessive based upon the pre-operative images provided.
More information about the indications for anterior plate supplemental fixation are needed. when was a plate used and not used? What were the criteria. Were fixed or variable angle screws used? Was bicortical purchase a goal with the screws?
Reviewer 2 Report
The study deals with the surgical treatment of pathological findings of the cervical spine treated by corpectomy and implantation of an expandable cage as a vertebral body replacement.
In degenerative or inflammatory pathologies in the area of the cervical spine (e.g. spinal canal stenosis or spondylodisics), a corpectomy may be indicated in particular with multisegmental compression or kyphotic malpositions. For this purpose, many different systems regarding the vertebral body replacement are available. In this paper, the expandable Cage X Core Mini® is examined in a retrospective data collection with regard to clinical results, differences in diagnosis, number of operated segments and subsidence. It is a retrospective study design with a case number of 86 patients who were surgically treated in the period between January 2012 and December 2019.
The explanations in the introduction are well written and logically structured, so that the questions raised are clearly justified. In the discussion, these questions are discussed in detail in the context of the current literature. The relevant current and relevant studies are presented in detail. Weaknesses of the present study, however, lie in the retrospective study design, the different pathologies of the surgically treated patients, different levels treated and the different follow-up period. On the positive side, it should be mentioned that the underlying patient collective has been divided into different groups, which also make the respective surgical procedure appear comprehensible to the reader. Further graphical representations would be helpful, e.g. various outcome parameters pre- and postoperatively plotted in the respective groups as a bar chart (VAS, Cage subsidence, EMS).
Regarding cage subsidence: Did you find any correlation with a special patient group (e.g. poor bone quality, spondylodiscitis…)? This would be very interesting.
Despite the retrospective data collection and the small number of patients, this study deals with a current and relevant topic.
Reviewer 3 Report
In this retrospective study the authors evaluate advantages & disadvantages of corpectomy reconstruction with an expandable cage 86 patients aged 61.3 years for a variety of indications in a 8 year period in a f/up of 30.7 months. Indications included spinal canal stenosis with myelopathy, metastasis , spondylodiscitis , and fracture. In 39 patients (45.3%) additional dorsal stabilization (360⁰ fusion) was performed. The authors concluded that the expandable titanium cages are a safe and useful tool in anterior cervical corpectomies for providing adequate anterior column support and stability.
I congratulate the authors for nice surgeries (attached figures)
Interesting issue. The authors have shown their work including all cases they operated with this expandable case. Correct diagnosis? What else surgeries they did with other devices or other approaches?
Disadvantages : Retrospective design, no control group, diversity of indications
Material & methods section is poor written with lack of substantial elements of methods. E.gt. clear indications, methods of stability evaluation and fusion .
In the Complications section/Results the authors have summed all complications mostly related with surgery, but more than three diagnoses were used for surgery in this paper. E.g. different issue in metastasis that in fracture!.
Some concerns regarding indications : E.g. In Figure 2 the authors have used anterior cage although the stenosis is >3 levels with excellent cervical lordosis. I would prefer to make a posterior approach for decompression instead.
One series e.g. degenerative disease plus one control group are the requirements for support the publicartion of this paper.
Round 2
Reviewer 3 Report
The authors have made much work to adres all my comments and actually did nice job adressing adequately. The only issue are the indications for surgey and non-homogenous sample of the patients operated .
Author Response
Please see the attached document
Response to Reviewer #3 Comments, Round 2:
Point 1.: The authors have made much work to adress all my comments and actually did nice job adressing adequately. The only issue are the indications for surgery and non-homogenous sample of the patients operated.
Response 1: We appreciate very much Reviewer #3 this suggestion for improvement. In the revised Methods and Materials Section we have described the indications for the single and multiple level surgery in different pathologies (Page 2 and 3, Line 49-98). The point on influence of different indications for surgery as well as non-homogenous sample of the patients operated on our results has been addressed in the Discussion Section on disadvantages and limitations of the study (Page 13, Line 364-376) as well as in the Conclusion section (Page 13, Line 381-382).
